# Community led testing among people who inject drugs: A community centered model to find new cases of HIV and Hepatitis C in Nepal

Rajesh Didiya[1], Deepak Gyenwali[1], Tara Nath Pokhrel[2], Sudha Devkota[2], Amrit Bikram Rai[1], Purusotam Raj Shedain[2], Mukunda Sharma[3], Madan Kumar Shrestha[2], Muhammad Imran[4], Zakir Kadirov[5], Bhawani Prasad Dahal[5], Prawchan K. C.[6], Sabir Ojha[1], Khem Narayan Pokhrel[1]*

1 National Association of People Living with HIV/AIDS in Nepal (NAP+N), Kathmandu, Bagmati, Nepal, 2 National Centre for AIDS and STD Control (NCASC), Kathmandu, Bagmati, Nepal, 3 National Public Health Laboratory (NPHL), Kathmandu, Bagmati, Nepal, 4 Save the Children, Fairfield, Connecticut, United States of America, 5 Save the Children, Nepal Country Office, KPRA Project, Kathmandu, Bagmati, Nepal, 6 SPARSHA Nepal, Lalitpur, Bagmati, Nepal

☯ These authors contributed equally to this work.

* pratikjee@gmail.com

**Data Availability Statement:** Data will be available upon request. 1. National Center for AIDS and STDs Control (Government body of Nepal) will be

## Abstract

### Background

People who inject drugs (PWIDs) have sub-optimal HIV and HCV testing as the available testing services are inadequate in low and middle-income countries. We examined a model of Community-Led Testing (CLT) in Nepal, exploring the feasibility of HIV and HCV testing by trained lay service providers who had similar backgrounds to those of PWIDs. We also assessed the prevalence of HIV and HCV within this study population and the associated risk factors among PWIDs.

### Methods

A mix-methods cross-sectional study was conducted among 1029 PWIDs in five major districts of Nepal from July 2019 to February 2020. Trained PWID peers performed the screening for HIV and HCV using Rapid Diagnostic Test (RDT) kits. Acceptability and feasibility of the testing was assessed. The participants' sociodemographic characteristics and injecting and non-injecting risk characteristics were determined. The association of risk and prevention characteristics with testing results were assessed using multiple logistic regression.

### Results

PWIDs shared that the test providers were friendly and competent in counseling and testing. Of total PWIDs (n = 1,029), 20.6% were HCV-positive and 0.2% were HIV-positive. HCV positivity was associated with needle sharing (AOR: 1.83, 95% CI: 1.27,2.64; p = 0.001) and reuse of syringe/needle (AOR: 2.26; 95% CI: 1.34, 3.79; p = 0.002). In addition, PWIDs were more likely to be HCV-positive who started opioid substitution therapy (OST) (AOR:

responsible for providing the data. Email:
ncasc@ncasc.gov.np 2. National Association of
People Living with HIV/AIDS in Nepal info@napn.
org.np.

**Funding:** The study is funded by "Global Fund to
Fight AIDS, Tuberculosis and Malaria for KPRA
Project".

**Competing interests:** The authors have declared
that no competing interests exist.

1.88, 95% CI: 1.26, 2.80, p = 0.002) and attended the rehabilitation center (AOR: 1.66, 95%
CI: 1.10, 2.53, p = 0.017).

## Conclusions

This CLT model was found to be a novel approach of testing of HIV and HCV which was
acceptable to PWIDs in Nepal and showed the high prevalence of HCV and its association
with injecting-related risk behaviors and being users of OST and rehabilitation. The findings
highlight the need of community-led testing in hotspots, OST settings, and rehabilitation
centers to screen new HIV and HCV infections.

## Background

People who inject drugs (PWIDs) are a high-risk key population for contracting human
immunodeficiency virus (HIV) and Hepatitis C virus (HCV) infections [1–3]. This group
accounts for the major proportion of HCV infections, globally [3]. An estimated 2.3 million
people living with HIV are co-infected with HCV globally [3, 4]. HCV-induced liver disease
has emerged as a major contributor to mortality and morbidity among people living with HIV
and are co-infected with HCV [5].

Current HIV and HCV testing services are inadequate in reaching the hard-to-reach high
risk key populations, particularly PWIDs, which has resulted in the under diagnosis of HIV
and HCV in low-and middle-income countries (LMICs) [6]. Such gaps in testing may require
adopting alternative approaches and strategies to achieve the target for elimination of HIV and
hepatitis C as public health threats by 2030 [6, 7].

Sero-prevalence studies conducted across various sites in Nepal have shown that HCV
prevalence among PWIDs is ranging from 21–38% in men and 3% in women in 2017. The
HIV/HCV co-infection was 2.5 to 7.4% in male PWIDs and 0.6% in female PWIDs depending
upon the study site in the country [8–10]. Although Nepal has made progress in improving
accessibility of testing and treatment of HIV, an estimated 22% of the HIV-positive people
remain undiagnosed [11]. In addition, HCV testing services are only available in clinic-based
settings, and PWIDs may not feel comfortable to seeking these testing services from the facili-
ties [12]. PWIDs also face social and environmental challenges such as poverty, homelessness,
criminalization of drug injecting, lack of confidentiality, stigma, and discrimination [13, 14].

In order to overcome the prevailing barriers and bridge the gap in testing of HIV and HCV,
an innovative model of Community-Led Testing (CLT) was conceived, developed, and tested
in Nepal. In this model, trained In-reach Workers (IRWs) who had similar backgrounds to
those of PWIDs performed HIV and HCV screening using easy to use rapid diagnostic test
kits (RDTs) in the community, bringing testing services closer to those most at risk and who
might not otherwise have been tested [15]. IRWs were the past injecting drug users and cur-
rently recovered from the drug use. They were also working as peer educators and lay service
providers to PWIDs through their engagement in community service organizations (CSOs) in
Nepal. Community led or community driven approaches have increased acceptability and
effectiveness of the interventions employed to improve access to HIV services in some African
countries [16–18]. Such a community-led model was also found to be effective in overcoming
barriers and reaching the hard to reach, vulnerable and stigmatized key populations in HIV
testing in Vietnam, and community home-based care support has improved ART adherence
in Nepal as well [19, 20]. CLT model for HIV screening has been implemented in Nepal since

2018. However, its acceptability and effectiveness is yet to be examined [15]. Moreover, there are no PWIDs focused HCV screening programs in community settings.

This study aimed to examine the acceptability and feasibility of a CLT model of testing for HIV and HCV among PWIDs. In addition, we assessed the prevalence of HIV, HCV, HIV/ HCV co-infection and the associated risk factors with HCV.

## Materials and methods

### Study design

A mix-methods cross-sectional CLT study was designed and conducted among PWIDs in five major districts of Nepal, namely Kathmandu, Lalitpur, Bhaktapur, Kaski, and Chitwan. The qualitative approach explored in-depth knowledge on acceptability and feasibility of this CLT model from clients' and providers' perspectives. While the quantitative approach was used to examine the risk characteristics of PWIDs and to measure the prevalence of HIV and HCV, the study spanned from July 2019 to February 2020.

### Study setting, sample size and sampling strategy

We purposively selected five districts where an estimated 30% of all PWIDs in Nepal reside [11]. Calculation of sample size was based on the current highest prevalence of HCV at 38% according to the integrated biological and behavioural survey (IBBS) conducted in Nepal [8]. Sample size was calculated using G*Power 3.1.9.6 for macOS, employing the 95% confidence interval and 80% power, the calculated sample size was 980. In addition to taking the reference prevalence of HCV, injecting related characteristics and their association with HCV positivity might need to be considered. We also aimed to identify new cases of HIV in the study areas and purposefully increased the sample size to 1,029. We applied convenience sampling methods to reach the PWIDs peers through IRWs.

### Data collection

Participants were recruited from satellite sites covering hot spots (e.g., the location of secret places where PWIDs generally meet to inject drugs), drop-in centres, needle/syringe exchange program locations, opioid substitution therapy (OST) sites, and rehabilitation centres. IRWs of the district-based community service organizations were trained with five-days intensive training package for conducting CLTs, counselling, and data collection including the practices on KOBO data collection platform using Android tablets [21]. Geographical tracking was also done to trace location of the participants from the office to the site where they were interviewed. Trained IRWs and peers were mobilized to identify and contact the participants.

For qualitative data collection, in-depth interviews (IDIs) were conducted by two experienced qualitative researchers covering 70 participants from five study districts who were enrolled in CLT for screening HIV and HCV. The researchers received 3 days training covering using topic guides, selecting participants, maintaining ethical procedures, recording, and transcribing the information. Similarly, three Focus Group Discussions (FGDs) were conducted in three project sites among key CLT service providers covering IRWs, lab personnel, and counsellors. Trained professionals were mobilized to conduct IDIs and FGD sessions. We used topic guides for IDIs and FGDs after conducting validation testing with the government and non-government stakeholders including experts in the study.

## Participant inclusion criteria

PWIDs were considered eligible for the study if they were current or past injecting drug users who, had injected drugs for at least 3 months before they were recruited in the study, and had never tested or tested negative for HIV and /or HCV within the 12 months preceding the survey. Participants aged 16 years and above were enrolled. All the participants meeting inclusion criteria had provided consent to enrol in the study and to provide information. Therefore, no participants from the sample were excluded in the study.

## Study variables

**Socio-demographic information.** Participants' age, ethnicity, educational status, marital status, gender, and occupation, history of HIV and HCV testing were recorded adopting the survey instruments of the Nepal Demographic Health Survey, 2016 [22].

**Injection related risk characteristics.** We adopted the questionnaires about the injection related risk characteristics covering needle sharing behaviors. Those who are injecting drugs less than three months were current drug users and those who injected drugs prior to three months are pas drug users. and other equipment sharing as defined by the Integrated biological and behavioral surveillance survey (IBBS) study in Nepal [8].

**HIV and HCV risk factors.** The risk factors were adopted from the Nepal IBBS study which assessed the burden of risk factors covering tattoos or body piercing, having sexual partner with status of HIV or with chronic hepatitis C/HIV, condom use, and weekly alcohol use [9].

**Prevention activities.** We interviewed the participants about their exposure to peer-education interventions, opioid substitution therapy, rehabilitation, needle and syringe exchange programs following the harm reduction programs activities of National Center for AIDS and STDs Control (NCASC). We applied both qualitative and quantitative approaches for assessment of prevention activities.

**Biological test.** As shown in the CLT algorithm (Fig 1), the eligible PWIDs from the satellite sites (hotspots, OST clinics, drop-in centers, and rehabilitation centers) were enrolled in the study. Following the proper pre-test counselling they were screened for HIV and HCV. HCV and HIV screening used RDTs. Alere Determine™ HIV-1/2 and SD BIOLINE HCV RDTs were used for screening HIV and HCV, respectively, using blood samples from finger pricks in the satellite sites covering various hotspots, OST clinics, drop-in centers, and rehabilitation centers. Trained and certified IRWs performed screening of HIV and HCV of PWIDs and recorded clients' information including behavioural charactersitics and test results in client report form (CRF) at satellite sites. PWIDs with non-reactive test results for HIV and HCV were counselled for safer behaviour and referred to harm reduction programs. PWIDs with reactive test results for HIV and /or HCV were accompanied to demonstration sites where confirmatory test of HIV was performed according to national HIV testing protocol using tests Uni-Gold HIV ½ and Stat pak HIV-1/2 [15]. The confirmed HIV positives were linked to treatment and care services after post-test counselling.

## Data analysis

Quantitative data were analyzed using STATA version 14.0. Descriptive analyses were done covering client's characteristics and the prevalence of HIV, HCV and HIV/HCV co-infection stratified by gender. We used chi-squared test to see the gender differences in risk characteristics, sociodemographic characteristics, and characteristics according to the HCV-positive and HCV-positive results. We applied t-test for continuous variables. Logistic regression analysis was applied to identify the association of socio-demographic characteristic and risk related

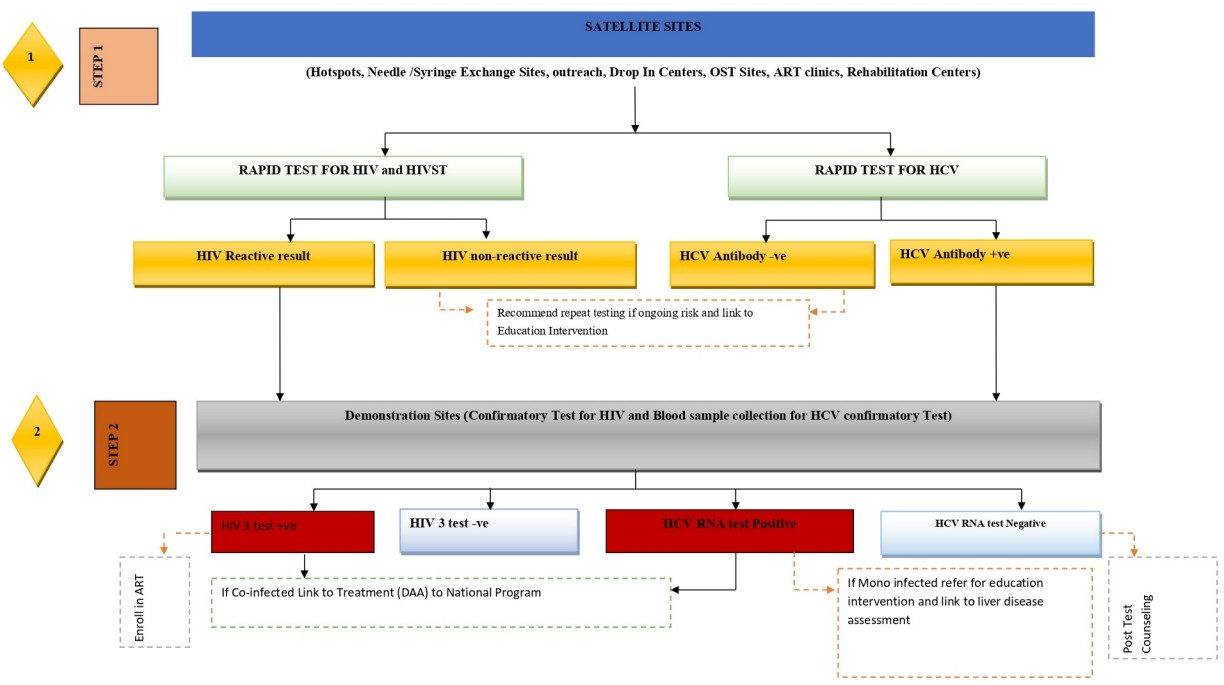

**Fig 1. The HIV/HCV CLT model in Nepal.**

characteristics with the prevalence of HCV. We applied univariate analysis and developed model covering age, sex, marital status, education, risk characteristics, prevention characteristics, and rehabilitation characteristics as these are considered the important risk factors for HCV results in Nepal. Results of HIV testing precluded meaningful analyses, since the number was so low.

The qualitative interviews and focus groups were transcribed and translated into English. The contents of transcripts were analyzed based on the pre-defined and emerging themes and sub-themes. We used MAXQDA 2020 software [23] for data coding, data sorting and extracting quotes according to theme.

## Ethical consideration

As PWIDs are vulnerable, socially stigmatized and criminalized for drug use in Nepal, the study was conducted maintaining optimal ethical and human rights standards throughout the study period. Written infomed consent was taken from all the participants before enrolling them in the study. Similarly, written assent was taken from senior peers of the participants below 18 years of age. Ethical approval was obtained from ethical review board of the Nepal Health Research Council (NHRC) [**Reg No. 552/2019**] prior to the commencement of the fieldwork.

Participants, who tested as HIV and HCV negative were counselled on safer behaviors and referred to harm reduction programs. Participants identified as either HIV and/or HCV positive were referred to the demonstration sites for counseling, HIV confirmatory test, linkages for HIV treatment, care and support, and to collect blood sample for HCV RNA/PCR. Participants identified as HCV positive were linked to post-test counselling services, liver fibrosis assessment through APRI scoring and referred to a hepatologist for appropriate follow up.

## Results

### Sociodemographic characteristics of participants

Of total participants (n = 1,029), 91.6% were men and 8.4% were women. Mean ages were 28.6 (SD = 7.0) years for men and 23.4 (SD = 3.7) years for women. Proportion of women in the age group of 16–24 years was 66.3% and that of men was 34.3%. The majority of participants were Janajati (Disadvantaged Janajati (Janajati other than Newar, Gurung, and Thakali): 26.3% and Advantaged Janajati (Newar, Gurung, and Thakali): 35.6%). About 63.1% of men and 75.6% of women were unmarried. Nearly two thirds (61.7%) of the participants were unemployed. Those who had ever enrolled in school, their mean year of schooling was 9.8 (SD = 2.4) (Table 1).

### Risk related characteristics of participants by gender

Of total (n = 1029), 56.8% were current drug users. More than half (51.6%) of the participants reported that they had never tested for HIV. Regarding risk behaviors, 75.0% reported that they reused needles, 50.5% shared needles/syringes, and 29.9% shared a cooker/vial/container. About 55% of the participants were recruited from hotspots and 29% were from rehabilitation centers.

About 61.0% of the participants had an injection history of more than five years, and 43.6% reported that they had injected in the last seven days. Regarding needle/syringe sharing, 50% of men and 57.0% of women reported that they shared needles. In addition, 29.9% of participants reported that they had a practice of sharing a cooker/vial/container. Regarding other risk characteristics, 30.4% of men and 5.8% of the women consumed alcohol weekly, and 57.0% of the total participants had done tattooing. Nearly half of the women (47.7%) responded that they did not know their partner's HIV status, and 37.5% of men did not know their partner's status. While having sex with regular partner/sex workers, 57.1% reported that they always used condoms (Table 2).

### Past testing, prevention exposure according to gender

Of the total, 17% of the participants reported that they had HCV test at least 12 months prior to the survey. Regarding the reason for not testing for HCV, 43.0% reported fear of confidentiality breaches and 39.5% were not aware of the risk of HCV and HIV. Other reasons were screening is too expensive (23.5%), carelessness within themselves (18.0%), and no treatment is available (11.5%). About 10.5% reported that they did not know the testing facilities. About 52.9% of men and 62.8% of women participated in the IRW provided harm reduction intervention, 28.1% started OST, and 63.2% attended a rehabilitation center. About 69.6% reported that they had participated in needle/syringe exchange program. About 61.8% reported that they felt stigma from family; a higher proportion of men compared to women (63.7% vs. 40.7%) felt stigma from family.

The overall prevalence of HCV in this sample was 20.6% (men = 21.2%; women = 14.0%). All HCV-positive participants were referred to demonstration sites to collect blood samples and the samples were sent to the National Public Health Laboratory [NPHL] for HCV RNA/ PCR test. Only two men (0.2%) tested HIV positive and had HCV positive test as well. HIV-positive samples were sent for confirmatory tests using three RDTs as per national HIV testing algorithm. Both participants were confirmed as HIV positive, and both were linked to ART services. Among those who tested for HCV RNA (n = 203), 68% had HCV RNA (IU/ml) count ranging from 65 to 1000000 (Table 3).

**Table 1. Socio-demographic characteristics of participants by gender.**

| Characteristics | Men (n = 943) (n, %) | Women (n = 86) (n, %) | Total (n, %) |
|---|---|---|---|
| Age (mean, SD) | 28.6 (7.0) | 23.4 (3.7) | 28.2 (7.0) |
| **Age groups** | | | |
| 16–24 years | 323 (34.3) | 57 (66.3) | 380 (36.9) |
| 25–50 years | 610 (64.7) | 29 (33.7) | 639 (62.1) |
| Above 50 years | 10 (1.0) | 0 (0.0) | 10 (1.0) |
| **Ethnicity** | | | |
| Dalit | 81 (8.6) | 8 (9.4) | 89 (8.7) |
| Disadvantaged Janajati | 245 (26.0) | 25 (29.0) | 270 (26.3) |
| Disadvantaged non-Dalit Terai | 2 (0.2) | 1 (1.2) | 3 (0.3) |
| Religious minorities (Muslim) | 8 (0.9) | 0 (0.0) | 8 (0.8) |
| Advantaged Janajati | 340 (36.1) | 26 (30.2) | 366 (35.6) |
| Upper caste (Brahmin/Chhetri) | 266 (28.2) | 26 (30.2) | 292 (28.3) |
| **Marital status** | | | |
| Unmarried | 595 (63.1) | 65 (75.6) | 660 (64.1) |
| Married | 339 (36.0) | 21 (24.4) | 360 (35.0) |
| Divorced/separated | 9 (1.0) | 0 (0.0) | 9 (0.9) |
| **Occupation** | | | |
| Unemployed | 562 (59.6) | 73 (84.9) | 635 (61.7) |
| Professional/technical/Managerial | 7 (0.7) | 0 (0.0) | 7 (0.7) |
| Labor | 96 (10.2) | 2 (2.3) | 98 (9.5) |
| Business | 134 (14.2) | 6 (7.0) | 140 (13.6) |
| Agriculture | 8 (0.9) | 0 (0.0) | 8 (0.8) |
| Student | 52 (5.5) | 3 (3.5) | 55 (5.3) |
| Others | 84 (8.9) | 2 (2.3) | 86 (8.4) |
| **Education level** | | | |
| Illiterate | 0 (0.0) | 8 (0.9) | 8 (0.8) |
| Literate but not formal education | 12 (1.3) | 4 (4.7) | 16 (1.6) |
| Formal education received | 923 (97.9) | 82 (95.4) | 1005 (97.7) |
| **Mean (SD) schooling years (n = 1005)** | 9.4 (2.4) | 9.8 (2.4) | 9.8 (2.4) |

### Descriptive analysis of socio-demographic and risk characteristics and their association with HCV screening result

Significantly higher proportion of participants in the age group of 25–50 years were HCV positive (77.4%) compared with those in the age group of 16–24 years and above 50 years (p<0.001). A higher proportion of married participants were HCV positive (57.1%) compared to unmarried and divorced/separated participants (p<0.001). HCV negative participants had a higher mean number of years of schooling than HCV positive [negative mean = 10.1 (SD = 2.2) vs. positive: mean = 8.9 (SD = 2.7), p<0.001].

A higher proportion of participants who reported needle sharing (58.0%) were HCV positive compared to those who did not share needles (p = 0.014). In addition, a higher proportion participant who shared a cooker/vial/container were HCV positive compared to those who did not share (positive: 37.3% vs. negative: 28.0%, p = 0.009). In addition, there was a significant difference on HCV positive and HCV negative results between those who reported re-use of needle/syringe compared with who did not reuse needle and syringes. (positive: 86.2% vs. negative: 72.1%, p<0.001).

**Table 2. Risk characteristics of participants by gender.**

| Characteristics | Men (n = 943, %) | Women (n = 86, %) | Total (n = 1029, %) |
|---|---|---|---|
| **Ever drug use** | | | |
| Current | 531 (56.3) | 54 (62.8) | 585 (56.8) |
| Past | 412 (43.7) | 32 (37.2) | 444 (43.2) |
| **Last test of HIV Done** | | | |
| Never | 490 (52.0) | 41 (47.7) | 531 (51.6) |
| Before 12 months | 453 (48.0) | 45 (52.3) | 498 (48.4) |
| **Risk behavior (multiple response)** | | | |
| Shared a needle | 471 (50.0) | 49 (57.0) | 520 (50.5) |
| Shared a cooker/vial/container | 288 (30.5) | 20 (23.3) | 308 (29.9) |
| Shared a cotton/filter/rinse water | 43 (4.6) | 1 (1.2) | 44 (4.3) |
| Re-used syringe/needle | 717 (76.0) | 55 (64.0) | 772 (75.0) |
| Not shared | 30 (3.2) | 0 (0.0) | 30 (2.9) |
| **Type of satellite** | | | |
| DIC | 33 (3.5) | 1 (1.2) | 34 (3.3) |
| Hotspots | 516 (54.7) | 50 (58.1) | 566 (55.0) |
| OST Clinic | 71 (7.5) | 2 (2.3) | 73 (7.1) |
| Rehabilitation center | 272 (28.8) | 31 (36.1) | 303 (29.5) |
| ART Clinic | 2 (0.2) | 0 (0.0) | 2 (0.2) |
| Others | 49 (5.2) | 2 (2.3) | 51 (5.0) |
| **Injecting duration** | | | |
| <= 1 year | 25 (2.7) | 7 (8.1) | 32 (3.1) |
| Between one year and five years | 323 (34.3) | 46 (53.5) | 369 (35.9) |
| More than five years | 595 (63.1) | 33 (38.4) | 628 (61.0) |
| **Last injected (days)** | | | |
| <= 7 days | 417 (44.2) | 32 (37.2) | 449 (43.6) |
| 8–30 days | 109 (11.6) | 22 (22.6) | 131 (12.7) |
| 31 days or more | 417 (44.2) | 32 (37.2) | 449 (43.7) |
| **Weekly alcohol use** | | | |
| Yes | 287 (30.4) | 5 (5.8) | 292 (28.3) |
| No | 656 (69.6) | 81 (94.2) | 737 (71.6) |
| **Tattoo** | | | |
| Yes | 546 (57.9) | 40 (46.5) | 586 (57.0) |
| No | 397 (42.1) | 46 (53.5) | 443 (43.0) |
| **Partner status** | | | |
| HIV-positive | 2 (0.2) | 0 (0.0) | 2 (0.2) |
| Hepatitis C | 2 (0.2) | 1 (1.2) | 3 (0.3) |
| Don't know | 354 (37.5) | 41 (47.7) | 395 (38.4) |
| None | 585 (62.0) | 44 (51.2) | 629 (61.1) |
| **Condom use with casual partners/SWs** | | | |
| Never | | | 51 (5.0) |
| Always | | | 588 (57.1) |
| Sometimes | | | 326 (31.7) |
| Never had sex with SWs | | | 64 (6.2) |

**Table 3. Past testing, prevention, stigma, and current testing results by gender.**

| Characteristics | Men (n = 943,%) | Women (n = 86,%) | Total (n = 1029) |
|---|---|---|---|
| **Past HCV test** | | | |
| Yes | 170 (18.0) | 5 (5.8) | 175 (17.0) |
| No | 773 (82.0) | 81 (94.2) | 854 (83.0) |
| **Reason for not testing HCV (multiple response)** | | | |
| Not aware of the risk | 299 (38.6) | 41 (50.6) | 340 (39.7) |
| Don't know about the testing facilities | 87 (11.2) | 3 (3.7) | 90 (10.5) |
| Testing facility is too far | 18 (2.3) | 1 (1.2) | 19 (2.2) |
| Fear of stigma in the facility | 60 (7.7) | 5 (6.2) | 65 (7.6) |
| Fear of confidentiality | 338 (43.6) | 30 (37.0) | 368 (43.0) |
| Screening is too expensive | 189 (24.4) | 12 (14.8) | 201 (23.5) |
| No treatment is available | 92 (11.9) | 6 (7.4) | 98 (11.5) |
| Carelessness | 148 (10.7) | 6 (0.7) | 154 (18.0) |
| **Participated in IRW intervention** | | | |
| Yes | 499 (52.9) | 54 (62.8) | 553 (53.7) |
| No | 444 (47.1) | 32 (37.2) | 476 (46.3) |
| **OST started** | | | |
| Yes | 278 (29.5) | 11 (12.8) | 289 (28.1) |
| No | 664 (70.5) | 75 (87.2) | 739 (71.9) |
| **Attended in Rehabilitation center** | | | |
| Yes | 600 (63.6) | 50 (58.1) | 650 (63.2) |
| No | 343 (36.8) | 36 (41.9) | 379 (36.8) |
| **Participated in needle/syringe exchange program** | | | |
| Yes | 653 (69.3) | 63 (73.2) | 716 (69.6) |
| No | 290 (30.8) | 23 (26.7) | 313 (30.4) |
| **Stigma** | | | |
| Stigma from family | 601 (63.7) | 35 (40.7) | 636 (61.8) |
| Stigma in the health facility | 64 (6.8) | 1 (1.2) | 65 (6.3) |
| Stigma in office/work | 42 (4.5) | 0 (0.0) | 42 (4.1) |
| Not felt stigma | 325 (34.5) | 51 (59.3) | 376 (36.5) |
| Stigma felt in other places | 38 (4.0) | 1 (1.2) | 39 (3.8) |
| **HCV screening result** | | | |
| Positive | 200 (21.2) | 12 (14.0) | 212 (20.6) |
| Negative | 743 (78.8) | 74 (86.0) | 817 (79.4) |
| **HIV screening result** | | | |
| Positive | 2 (0.2) | 0(0.0) | 2 (0.2) |
| Negative | 941 (99.8) | 86 (100.0) | 1027 (99.8) |
| **HIV confirmatory test 3RDT proposed (yes)**[as per national HIV testing algorithm] | 2 (100.0) | 0 (0.0) | 2 (100.0) |
| **HCV RNA (IU/ml) (n = 203)** | | | |
| <65 | | | 61 (30.0) |
| 65 to 1000000 | | | 138 (68.0) |
| >1000000 | | | 4 (2.0) |

Regarding risky sexual behaviors, those participants who had sexual contact with a person of unknown HIV status had higher proportion of HCV positive results (positive: 30.2% vs. negative: 20.3%, p = 0.002) compared with participants who had no sexual contact or knew the HIV status of their partner. Those participants who always used condoms had a low proportion of HCV positive results compared to those who did not use condoms and used them only

sometimes (positive: 37.4% vs. negative: 50.6%, p = 0.001). Also, PWIDs with tattoos had a higher prevalence of HCV positive than those without tattoos(positive: 67.5% vs. negative: 54.2%, p<0.001). Similarly, participants who consumed alcohol weekly had a higher proportion of HCV positive results relative to those who did not consume alcohol weekly (positive: 75.9% vs. negative: 67.9%, p = 0.024).

A higher proportion of HCV positive PWIDs had started OST compared to HCV negative (positive: 46.2% vs. negative: 23.4%, p<0.001). Rehabilitation participants had a higher proportion of HCV positive test results (positive: 76.4% vs. negative: 59.7%, p<0.001) relative to non-participants, and a higher proportion of HCV positive results were found among subjects who were involved in needle/syringe exchange programs (positive: 75.9% vs. negative: 67.9%, p = 0.024) (Table 4).

## Multivariate analysis of injecting risk behavior, prevention exposure associated with HCV screening results

Analyses were performed controlling for age, sex, marital, education level, tattooing, and participation in a needle/ syringe exchange program. Higher odds of HCV positive results were found in participants who had practice of needle sharing (AOR: 1.83, 95% CI: 1.27,2.64, p = 0.001) and reuse of syringes/needles (AOR: 2.26, 95% CI: 1.34, 3.79, p = 0.002). In addition, HCV positive results were positively associated with those who started OST (AOR: 1.88, 95% CI: 1.26, 2.80, p = 0.002) and attended a rehabilitation center (AOR: 1.66, 95% CI: 1.10, 2.53, p = 0.017.) (Table 5).

## Qualitative findings

**Feasibility of CLT.**   IRWs shared their experience and perceptions about the feasibility of community led testing. Doing RDT was convenient for them because they found that PWIDs were ready to get tested and agreed to be tested by their peers. They shared the positive and encouraging aspect of this model was "testing by the community for the community." The trained IRWs shared that they developed confidence as they had proper skill for procedure through the training and having received a certificate from a government authority. Another aspect shared by the trained peer was that they felt empowered and employed to test their community for the benefit of the community. The testing was easier for them as they found willingness among the peer PWIDs to get tested and refer their friends for testing for HIV and HCV.

*"I never thought that I would have this much skills in counseling and testing. The training made me more skilled in testing procedures. I felt myself proud to reach my peer PWIDs who never get tested. It is really the rewarding job testing by the community for the community"*

- - -26 years IRW, Pokhara.

**Acceptability of CLT.**   The community led testing was acceptable among the participants as they reported that they found friendly behavior among test providers and perceived that the trained test providers had competence in counseling and testing. In addition, participants reported that they were happy to have immediate test results and having the testing place of their choice. Regarding privacy and confidentiality, PWIDs felt assured about the privacy maintained by the IRWs for testing. They had also greater trust toward the interviewers and test takers in the field. They had trust on pre-counselling and testing, and felt that their community are the service provider for the benefit of their diagnosis. Their perception was that they received exceptional support from their peers in testing and counseling from trained IRWs.

**Table 4. Descriptive analysis of socio-demographic and risk characteristics on HCV screening results.**

| Characteristics | HCV positive (n = 212, %) | HCV negative (n = 817, %) | p-value |
|---|---|---|---|
| **Age groups** | | | <0.001 |
| 16–24 years | 40 (18.9) | 340 (41.6) | |
| 25–50 years | 164 (77.4) | 475 (58.2) | |
| Above 50 years | 8 (3.8) | 2 (0.2) | |
| **Gender** | | | 0.111 |
| Men | 200 (94.3) | 743 (90.9) | |
| Women | 12 (5.7) | 74 (9.1) | |
| **Marital status** | | | <0.001 |
| Unmarried | 88 (41.5) | 572 (70.0) | |
| Married | 121 (57.1) | 239 (29.3) | |
| Divorced/separated | 3 (1.4) | 6 (0.7) | |
| **Mean (SD) schooling years (n = 1005)** | 8.9 (2.7) | 10.1 (2.2) | <0.001 |
| **Needle Sharing** | | | 0.014 |
| Yes | 123 (58.0) | 397 (48.6) | |
| No | 89 (42.0) | 420 (51.4) | |
| **Shared a cooker/vial/container** | | | 0.009 |
| Yes | 79 (37.3) | 229 (28.0) | |
| No | 133 (62.7) | 588 (71.9) | |
| **Reused needle/syringe** | | | <0.001 |
| Yes | 183 (86.2) | 589 (72.1) | |
| No | 29 (13.7) | 228 (27.9) | |
| **Sexual contact with person with HIV unknown status (yes)** | | | 0.002 |
| Yes | 64 (30.2) | 165 (20.3) | |
| No | 148 (69.8) | 647 (79.7) | |
| **Condom use** | | | 0.001 |
| No | 44 (20.9) | 162 (19.6) | |
| Always | 79 (37.4) | 413 (50.6) | |
| Sometimes | 88 (41.7) | 241 (29.5) | |
| **Tattoo done** | | | 0.001 |
| No | 69 (32.6) | 374 (45.8) | |
| Yes | 143 (67.5) | 443 (54.2) | |
| **Weekly alcohol consumption** | | | 0.024 |
| No | 51 (24.0) | 262 (32.1) | |
| Yes | 161 (75.9) | 555 (67.9) | |
| **Participated in peer education/IRW intervention** | | | 0.869 |
| No | 97 (45.8) | 379 (46.4) | |
| Yes | 115 (54.3) | 438 (53.6) | |
| **Started OST** | | | <0.001 |
| No | 114 (53.8) | 625 (76.6) | |
| Yes | 98 (46.2) | 191 (23.4) | |
| **Attended rehabilitation center** | | | <0.001 |
| No | 50 (23.6) | 329 (40.3) | |
| Yes | 162 (76.4) | 488 (59.7) | |
| **Participated in needle/syringe exchange program** | | | 0.024 |
| No | 51 (24.1) | 262 (32.1) | |
| Yes | 161 (75.9) | 555 (67.9) | |

**Table 5. Multivariate analysis of injecting risk behavior, prevention exposure with HCV screening results.**

| Characteristics | Unadjusted odds Ratio | 95% CI | p-value | Adjusted odds Ratio[1] | 95% CI | P-Value |
|---|---|---|---|---|---|---|
| Needle sharing (yes) | 2.46 | 2.02, 2.98 | 0.001 | 1.83 | 1.27, 2.64 | 0.001 |
| Reused syringe/needle (yes) | 2.52 | 2.11, 3.09 | 0.009 | 2.26 | 1.34, 3.79 | 0.002 |
| Currently taking OST (yes) | 2.80 | 2.05, 3.85 | <0.001 | 1.88 | 1.26, 2.80 | 0.002 |
| Attended Rehabilitation Center (yes) | 2.18 | 1.54, 3.08 | <0.001 | 1.66 | 1.10, 2.53 | 0.017 |
| Participated needle syringe exchange program (yes) | 1.49 | 1.05, 2.10 | 0.024 | 0.67 | 0.42, 1.08 | 0.108 |
| Tattooing (yes) | 1.74 | 1.27, 2.40 | 0.001 | 1.18 | 0.76, 1.82 | 0.455 |
| Age | 1.16 | 1.13, 1.19 | 0.001 | 1.10 | 1.05, 1.22 | <0.001 |
| Sex (male) | 1.65 | 0.88, 3.11 | 0.115 | 0.52 | 0.25, 1.11 | 0.093 |
| Marital status (married) | 1.20 | 0.89, 1.82 | 0.001 | 0.84 | 0.55, 1.22 | 0.427 |
| Mean year of schooling | 0.81 | 0.76, 0.86 | <0.001 | 0.79 | 0.73, 0.85 | <0.001 |

[1]Adjusted for age, sex, marital, mean year of schooling, tattooing, participated in needle syringe exchange program, needle sharing, need syringe/needle, OST started, attended rehabilitation center.

*"I got happy to meet my friend in the usual injecting site, where we used to be there together. He seemed to be very impressive. He provided me a pre-test counseling and testing at the spot in a confidential environment. I relieved to get the test results instantly".*

- 34-years PWID, Chitwan

**Bottlenecks/Challenges.** A few IRWs perceived that it was difficult for them in finding PWIDs who have not tested. Another problem that the trained peer said that in some cases they had difficulty in finding a private testing venue because some were feeling uncomfortable because police might interfere- as drug use is treated as a criminal act in Nepal. However, the majority of the IRWs reported that this incidence did not affect them in continuation of testing and finding new people for the testing. Some thought that they had problems convincing the peers about the treatment available for HCV infection if they tested positive. However, they referred positive participants for the consultation with the hepatologist.

*"I feel sometimes difficult to overcome the interference of police although our protocol says we can test in the hotspots. Once, I got embarrassed when we encountered the police in Dumbarahi site while testing. Nevertheless, I kept on continuing testing because I feel good to work for our community"*

- - -29 years IRW, Kathmandu

## Discussion

This study is the first of its kind in Nepal, which applied CLT model to screen the hidden PWIDs affected by HIV and HCV. The model was proven a feasible testing approach for HIV and HCV in a low-resource setting such as Nepal and was acceptable by PWIDs. The lay test providers and peers/IRWs had confidence in reaching, providing counseling, maintaining the privacy of PWIDs and testing in the satellite sites. The test providers also felt empowered as they were doing something for their own community: "Testing by the community for the community". Although a few service providers had some difficulties in finding private testing locations and felt fear of interference from the police, they were able to reach the PWIDs in convenient places and provide testing upon request. In addition, PWIDs shared that the test

providers were friendly and were competent in counseling and testing. They shared that the best part of the CLT was that they received the result instantly and were referred for further services. This model can be one of the approaches to overcome the access barriers for PWIDs for seeking services from clinical settings [14, 19, 24].

This study found that 0.2% of participants had HIV and 20% had HCV. The lower rate of HIV could be due to better utilization of testing services and their exposure to prevention services. HIV testing services are more accessible in the study areas and have better linkage to treatment and care. Also, those who had HIV testing prior to 12 months of the survey were not included in this model. The reliability of testing was promising as the result accuracy was 100% while performing confirmatory testing in National Public Health Laboratory (NPHL).

HCV positivity result is in line with the IBBS study findings, which has shown the prevalence ranging from 18.8 to 38%. The true prevalence of HCV among PWIDs could be even higher than that revealed by the study if the criteria "never tested or tested HCV negative at least for 1ast 12 months" were not used. This statement is supported by the findngs of a study conducted in 2015 which had shown almost 50% of the PWIDs with HCV-RNA sought services from organized institutions [25].

PWIDs represent 3% of the total cases of HIV in Nepal and HIV incidence was 0.03% in 2020 and the prevalence among adult population was 0.13% [11]. The study results indicated that PWIDs who were not reached by the ongoing program for HIV screening and could be reached through this peer-led approach [17, 19] although detection of HIV is fairly low. This CLT approach would be complementary to the current HIV/HCV screening services, as the PWIDs feel stigma and fear of confidentiality breaches in the health facilities that have resulted in non-availability or inaccessibility of the screening services [14, 19, 24].

This study also found that needle sharing and needle reuse were major risk factors for HCV infection. Reuse of needles used by oneself and received from others and needle/ syringe sharing may have increased the risk of vulnerability of transmission. In addition, HCV positivity rate was found to be high among PWIDs who started their OST and attended rehabilitation center. Participants seeking services from institutions may have history of years of injecting drug use before enrollment in the services and might have not used the available testing for HCV services with a fear of lack of treatment availability of HCV. Longer duration of exposure is positively associated with HCV infection and the longer duration increases the repeated high risk behaviors such as needle sharing, re-use of needles, and unsafe sexual behaviors [25, 26].

Despite the major significance of this CLT model, the study has two major limitations. First, due to nature of screening hidden PWIDs, we applied non-probability sampling led by peer PWIDs. This may affect the generalizability of the results. However, IRWs were reaching every satellite site with the mapping so that they would be able to reach more hidden and non-diagnosed cases. Second, we were not able to assess the factors associated with HIV-positivity because of the low prevalence in the screened populations. However, the risk characteristics of HCV and HIV could be assumed similar such as needle sharing and sharing of injecting preparation materials.

## Conclusion

This CLT study has shown that peer lay test providers were well accepted by PWIDs for counseling and testing. This model has become promising to test for HCV and HIV among PWIDs in their convenient sites. Also, the study shows that risk behaviour such as needle re-use and needle sharing led to HCV infection. Moreover, higher proportion of PWIDs in rehabilitation center and on OST were also tested positive for HCV. Therefore, this study

highlights the expansion of this model to screen for both HCV and HIV together in other parts of the country and similar settings, where PWIDs encounter barriers to access to testing services.

## Acknowledgments

We acknowledge the CSOs involved in this research project: SPARSHA Nepal-Kathmandu, Community Support Group-Kaski, and Nirnaya-Chitwan. We would like to thank the IRWs, who successfully conducted the testing and field work for quantitative data collection and qualitative researchers who conducted IDIs and FGDs. We owe to all 1029 PWIDs who successfully and willingly participated in this study, provided information, agreed for the test of HIV and HCV in the satellite places. We have also number of consultants to acknowledge who provided the inputs in design. Further, we would like to acknowledge the support received from staffs of Save the children, NCASC, NAP+N, and NPHL.

## Author Contributions

**Conceptualization:** Rajesh Didiya, Tara Nath Pokhrel, Amrit Bikram Rai, Zakir Kadirov, Bhawani Prasad Dahal, Khem Narayan Pokhrel.

**Data curation:** Deepak Gyenwali, Khem Narayan Pokhrel.

**Formal analysis:** Deepak Gyenwali, Khem Narayan Pokhrel.

**Investigation:** Deepak Gyenwali, Tara Nath Pokhrel, Zakir Kadirov, Bhawani Prasad Dahal.

**Methodology:** Deepak Gyenwali, Zakir Kadirov, Bhawani Prasad Dahal, Khem Narayan Pokhrel.

**Project administration:** Rajesh Didiya, Zakir Kadirov.

**Software:** Khem Narayan Pokhrel.

**Supervision:** Rajesh Didiya, Deepak Gyenwali, Tara Nath Pokhrel, Sudha Devkota, Zakir Kadirov, Bhawani Prasad Dahal.

**Validation:** Rajesh Didiya, Deepak Gyenwali, Tara Nath Pokhrel, Sudha Devkota, Purusotam Raj Shedain, Mukunda Sharma, Madan Kumar Shrestha, Zakir Kadirov, Bhawani Prasad Dahal, Khem Narayan Pokhrel.

**Writing – original draft:** Khem Narayan Pokhrel.

**Writing – review & editing:** Rajesh Didiya, Deepak Gyenwali, Tara Nath Pokhrel, Sudha Devkota, Amrit Bikram Rai, Purusotam Raj Shedain, Mukunda Sharma, Madan Kumar Shrestha, Muhammad Imran, Zakir Kadirov, Bhawani Prasad Dahal, Prawchan K. C., Sabir Ojha, Khem Narayan Pokhrel.

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
