## [Decision Letter · Decision Letter 0]

19 Apr 2021

PONE-D-21-04106

Community Led Testing among People Who Inject Drugs: A community centered model  to find new cases of HIV and Hepatitis C in Nepal

PLOS ONE

Dear Dr. Pokhrel,

Thank you for submitting your manuscript to PLOS ONE. After careful consideration, we feel that it has merit but does not fully meet PLOS ONE’s publication criteria as it currently stands. Therefore, we invite you to submit a revised version of the manuscript that addresses the points raised during the review process.

We look forward to receiving your revised manuscript.

Kind regards,

John S Lambert

Academic Editor

PLOS ONE

Journal Requirements:

3. When reporting the results of qualitative research, we suggest consulting the COREQ guidelines: http://intqhc.oxfordjournals.org/content/19/6/349. In this case, please consider including more information on the number of interviewers, their training and characteristics; and please provide the interview guide used.

[Further, we would like to acknowledge the support 414received fromtheadministrative, logistic, and financial staffs of Save the children, NCASC, 415NAP+N, and NPHL.]

 [The author(s) received no specific funding for this work.]

6. Please amend your list of authors on the manuscript to ensure that each author is linked to an affiliation. We note that affiliation 3 is not linked. Authors’ affiliations should reflect the institution where the work was done (if authors moved subsequently, you can also list the new affiliation stating “current affiliation:….” as necessary).

7. Please ensure that you refer to Figure 1 in your text as, if accepted, production will need this reference to link the reader to the figure.

Reviewers' comments:

Reviewer's Responses to Questions

**Comments to the Author**

1. Is the manuscript technically sound, and do the data support the conclusions?

Reviewer #1: Yes

Reviewer #2: Yes

2. Has the statistical analysis been performed appropriately and rigorously? 

Reviewer #1: Yes

Reviewer #2: Yes

3. Have the authors made all data underlying the findings in their manuscript fully available?

Reviewer #1: No

Reviewer #2: Yes

4. Is the manuscript presented in an intelligible fashion and written in standard English?

Reviewer #1: Yes

Reviewer #2: Yes

5. Review Comments to the Author

Reviewer #1: The manuscript describes an important contribution to the literature in Nepal - the role of peer led approaches to addressing HIV and HCV testing. While these data have been repeatedly shown in many other studies, it is important to show this is feasible in Nepal. Overall, the manuscript is well-written. However, there is lack of sufficient detains on the peer-led approach itself. In particular, there is a need for clarification of characteristics of IRWs, who are defined as having “similar backgrounds to PWID”. What characteristics are included to determine this similarity? Current/former PWIDs themselves? Similar age? The definition of peers needs more clarification in order to determine potentially what components might important to recommend for future interventions.

Other minor definitions/clarifications needed including:

-Figure 1 –coloring scheme makes it difficult to read. There is no text describing/summarizing what is going on in this figure. This would be helpful.

-Define advantaged vs. disadvantaged Janajati

-Define reused needles—does this include only individuals who resused needles but did not share it? The discussion statement—“resused of needles….may have increased the risk of …transmission” implies that these were also potentially shared? Please clarify as resusing a needle only used by oneself should not pose any transmission risk.

-Word choice of “proved” in the second sentence of the discussion—rephrase to “shown to be” or similar.

Reviewer #2: SUMMARY

In this mixed methods cross-sectional study, investigators assessed the feasibility and acceptability of a novel community-led testing intervention among PWID in five districts of Nepal. The authors also assessed the prevalence of HIV and HCV within the study population, as well as the associated risk factors for HCV. Overall, this work represents an important contribution to the literature. As the authors outline, PWID have historically been marginalized from HIV/HCV testing services, and the intervention outlined herein may be replicated to increase HIV/HCV screening in other LMICs. Some areas whether additional clarification would benefit the readers is outlined below. Importantly, it is unclear why sociodemographic characteristics; risk characteristics; and past testing, prevention, stigma, and current testing are stratified by gender rather than HCV screening status, which is a primary outcome of the analysis. Additional details and recommendations are outlined below.

ABSTRACT

- Line 33: “People who inject drugs” does not require capitalization

- Line 40: “mixed methods” (rather than mix-method) is the standard terminology for this approach

INTRO

- Lines 83-85: “In addition, HCV testing services are only available in clinic-based settings, and PWIDs may not feel comfortable to seeking these testing services from the facilities.” – While this is accurate, a citation would strengthen this statement.

- Lines 85-87: This statement would be strengthened by noting that these social/environmental challenges are barriers to HIV/HCV testing.

METHODS

- Lines 126-127: It is somewhat unclear how the final sample size of 1029 was determined. Was there a calculation to determine how many persons in excess of 980 were needed? Please clarify.

- Line 129: Typographical error. I believe this should read, “the location of secret places”

- There is quite a large number of uncommon acronyms in this section, and this makes the methodology a bit hard to follow. I would suggest using PWID, HIV, HCV, and CLT, (Maybe IBBS and IRWs) then spell out any of the remaining acronyms for clarify.

- Lines 141-142: “Validated IDIs and FGD topic guides were used in the study.” Are citations available for these validated measures?

- Lines 160-163: It is unclear whether these prevention activities were assessed qualitatively or quantitatively. Please specify.

- In the introduction, the authors note that the aims of the study were to “examine the acceptability and feasibility of a CLT model of testing for HIV and HCV among PWIDs and linking PWIDs to the available tratment [note spelling error: “treatment”], care, and support services.” However, the methods section lacks information about how linkage to treatment, care, and support services was measured and assessed. This is particularly unclear given the cross-sectional design of this study.

- There is no reference to Figure 1 in the text, though I assume that this should appear in the methods section.

- Please expand the data analysis section to detail which specific descriptive analyses were conducted.

- How were the final variables included in the multivariate analysis determined? Please specify.

RESULTS

- I would suggest moving Tables 1-3 to a supplemental file. While stratifying these data by gender is informative, stratification by HCV screening status (as is done in tables 4-5) more directly supports the aims of this study. It may also be informative to present some of the risk characteristics and/or past testing, prevention, stigma, etc. by HCV status; I will leave this determination to the authors.

- It is unclear why risk-related characteristics and past testing, etc. are stratified and presented by gender. This strikes me as a secondary analysis that would be more appropriate presented/summarized in a supplemental files. The authors may consider presenting a portion of these data stratified by HCV status in one additional primary table.

- Line 226: It is unclear how current vs. past drug use was defined.

- Risk related characteristics of participants by gender and Past testing, prevention exposure according to gender should be relocated to a supplemental file. It would be more informative to present some of these findings stratified by HCV status, as this is the primary outcome of interest in this study.

- Line 286: It is unclear how “started OST” was defined. For example, was this ever, in the past year, or current OST?

- Sections of the qualitative results should be reviewed for academic tone. For example, lines 340-341: “However, they never got discouraged and kept on going for testing and finding the community.” This sentence should be re-worked for clarity, informal terminology should be removed, and it should be framed based on what IRWs reported in interviews (e.g., “Some IRWs reported…”).

- The second and third quotes (page 13) are a bit difficult to follow. Is it possible for these translations to be enhanced?

DISCUSSION

- Line 346: This should read “This study is the first of its kind…”

- Lines 359-361: “The lower rate of HIV could be due to better utilization of testing services and their exposure to prevention services.” – Could the authors elaborate here? Are HIV testing services more accessible/available in these regions?

TABLES/FIGURES

- Figure 1 is difficult to discern; a higher-resolution image of this figure should be provided.

- Findings stratified by gender should be relocated to a supplemental file (see details above)

- In Table 6, the referent categories for all characteristics should be included. For example, “Needle sharing (yes)” and “Age (per year)”. Educational level should be binary with the referent category defined, unless mean schooling years was used instead.

- In the text (and possible in a note below the table) the authors should specify the recall period (e.g., needle sharing in the past year or ever?) and define “OST started” and “attended rehabilitation center” (i.e., are this ever, past year, or current?)

- Tables 4 and 5 could be collapsed into a single table.

- In Table 6, the authors might also consider presenting the unadjusted OR estimates

6. PLOS authors have the option to publish the peer review history of their article (what does this mean?). If published, this will include your full peer review and any attached files.

Reviewer #1: No

Reviewer #2: No

---

## [Author Response · Author response to Decision Letter 0]

12 May 2021

Thank you very much for providing valuable feedback. Please find the attached response sheet.

---

## [Editor Report · Decision Letter 1]

17 May 2021

Community Led Testing among People Who Inject Drugs: A community centered model  to find new cases of HIV and Hepatitis C in Nepal

PONE-D-21-04106R1

Dear Dr. Pokhrel,

We’re pleased to inform you that your manuscript has been judged scientifically suitable for publication and will be formally accepted for publication once it meets all outstanding technical requirements.

Kind regards,

John S Lambert

Academic Editor

PLOS ONE

Additional Editor Comments (optional):

all corrections made to satisfactory standard
---

## [Editor Report · Acceptance letter]

19 May 2021

PONE-D-21-04106R1 

Community led testing among people who inject drugs: A community centered model  to find new cases of HIV and Hepatitis C in Nepal 

Dear Dr. Pokhrel:

I'm pleased to inform you that your manuscript has been deemed suitable for publication in PLOS ONE. Congratulations! Your manuscript is now with our production department. 

Kind regards, 

on behalf of

Dr. John S Lambert 

Academic Editor

PLOS ONE